# The cargo receptor SURF4 promotes the efficient cellular secretion of PCSK9

Brian T Emmer[1,2], Geoffrey G Hesketh[3], Emilee Kotnik[2], Vi T Tang[2], Paul J Lascuna[2], Jie Xiang[2], Anne-Claude Gingras[3,4], Xiao-Wei Chen[2,5], David Ginsburg[1,2,6,7,8]*

[1]Department of Internal Medicine, University of Michigan, Ann Arbor, Michigan; [2]Life Sciences Institute, University of Michigan, Ann Arbor, Michigan; [3]Centre for Systems Biology, Lunenfeld-Tanenbaum Research Institute, Sinai Health System, Toronto, Canada; [4]Department of Molecular Genetics, University of Toronto, Toronto, Canada; [5]State Key Laboratory of Membrane Biology, Institute of Molecular Medicine, Center for Life Sciences, Peking University, Beijing, China; [6]Department of Human Genetics, University of Michigan, Ann Arbor, Michigan; [7]Department of Pediatrics and Communicable Diseases, University of Michigan, Ann Arbor, Michigan; [8]Howard Hughes Medical Institute, University of Michigan, Ann Arbor, Michigan

**Abstract** PCSK9 is a secreted protein that regulates plasma cholesterol levels and cardiovascular disease risk. Prior studies suggested the presence of an ER cargo receptor that recruits PCSK9 into the secretory pathway, but its identity has remained elusive. Here, we apply a novel approach that combines proximity-dependent biotinylation and proteomics together with genome-scale CRISPR screening to identify SURF4, a homologue of the yeast cargo receptor Erv29p, as a primary mediator of PCSK9 secretion in HEK293T cells. The functional contribution of SURF4 to PCSK9 secretion was confirmed with multiple independent *SURF4*-targeting sgRNAs, clonal SURF4-deficient cell lines, and functional rescue with *SURF4* cDNA. SURF4 was found to localize to the early secretory pathway where it physically interacts with PCSK9. Deletion of *SURF4* resulted in ER accumulation and decreased extracellular secretion of PCSK9. These findings support a model in which SURF4 functions as an ER cargo receptor mediating the efficient cellular secretion of PCSK9.

DOI: https://doi.org/10.7554/eLife.38839.001

*For correspondence:
ginsburg@umich.edu

## Introduction

PCSK9 is a proprotein convertase that acts as a negative regulator of the LDL receptor (*Seidah et al., 2014*). PCSK9 is synthesized primarily in hepatocytes and secreted into the bloodstream. Circulating PCSK9 binds to the LDL receptor and diverts it to lysosomes for degradation, thereby leading to decreased LDL receptor abundance at the hepatocyte cell surface, decreased LDL clearance, and hypercholesterolemia. PCSK9 was originally implicated in cardiovascular disease when human genetic studies identified gain-of-function PCSK9 mutations as a cause of familial hypercholesterolemia (*Abifadel et al., 2003*). Subsequently, loss-of-function PCSK9 variants were associated with decreased plasma cholesterol and lowered lifetime incidence of cardiovascular disease (*Cohen et al., 2006*; *Benn et al., 2010*). Therapeutic inhibitors of PCSK9 have been recently developed that exhibit potent lipid-lowering effects and are associated with a reduction in cardiovascular events (*Open-Label Study of Long-Term Evaluation against LDL Cholesterol (OSLER) Investigators et al., 2015*; *ODYSSEY LONG TERM Investigators et al., 2015*).

A critical early sorting step for secreted proteins is their incorporation into membrane-bound vesicles that transport cargoes from the ER to the Golgi apparatus (*Zanetti et al., 2011*). The formation of these vesicles is driven by coat protein complex II (COPII), which includes the SAR1 GTPase, heterodimers of SEC23/SEC24, and heterotetramers of SEC13/SEC31. Secreted cargoes are incorporated into COPII vesicles by two mechanisms. 'Cargo capture' refers to concentrative, receptor-mediated, active sorting of selected cargoes, in contrast to 'bulk flow', by which cargoes enter COPII vesicles through passive diffusion. These mechanisms are not mutually exclusive, as cargoes may exhibit basal rates of secretion that are enhanced by receptor-mediated recruitment. It remains unclear to what extent protein recruitment into the secretory pathway is driven by selective cargo capture versus passive bulk flow (*Barlowe and Helenius, 2016*).

The active sorting of secreted cargoes into COPII-coated vesicles is driven primarily by SEC24, with the multiple SEC24 paralogs observed in vertebrates thought to accommodate a diverse and regulated repertoire of cargoes. Genetic deficiency in the mouse for one of these paralogs, SEC24A, results in hypocholesterolemia due to reduced secretion of PCSK9 from hepatocytes (*Chen et al., 2013*). This finding suggested an active receptor-mediated mechanism for PCSK9 recruitment into COPII vesicles. A direct physical interaction between SEC24A and PCSK9, however, is implausible since SEC24A localizes to the cytoplasmic side of the ER membrane and PCSK9 to the luminal side, with neither possessing a transmembrane domain. This topology instead implies the presence of an ER cargo receptor, a transmembrane protein that could serve as an intermediary between the COPII coat and luminal PCSK9.

Although COPII-dependent ER cargo receptors were first identified in yeast nearly two decades ago, few examples of similar cargo receptor interactions have been reported for mammalian secreted proteins (*Barlowe and Helenius, 2016*). Previous investigation of the ER cargo receptor LMAN1 demonstrated no specificity for SEC24A over other SEC24 paralogs, making this unlikely to serve as a PCSK9 cargo receptor (*Wendeler et al., 2007*). Earlier analyses of PCSK9-interacting proteins (*Ly et al., 2016*; *Xu et al., 2012*; *Denis et al., 2011*) did not identify a clear receptor mediating PCSK9 secretion. Here, we developed a novel strategy for ER cargo receptor identification that combines proximity-dependent biotinylation with CRISPR-mediated functional genomic screening. This approach led to the identification of the ER cargo receptor SURF4 as a primary mediator of PCSK9 secretion in HEK293T cells.

## Results

### Identification of candidate PCSK9 cargo receptors by proximity-dependent biotinylation

To identify PCSK9-interacting proteins, we first engineered cells expressing a fusion of PCSK9 and a mutant biotin ligase, *E. coli* BirA*(R118G), that catalyzes proximity-dependent biotinylation of primary amines on neighboring proteins within an estimated ~10 nm radius (*Roux et al., 2012*; *Kim et al., 2014*), effectively converting transient interactions into covalent modification (*Figure 1A*). The high affinity of the biotin-streptavidin interaction in turn allows for stringent detergent and high salt conditions during purification. Quantitative mass spectrometry of streptavidin-purified interacting proteins from cells expressing PCSK9-BirA* identified 162 prey proteins that were specifically labeled (Bayesian FDR $\leq$ 1%) by PCSK9-BirA* relative to control bait proteins (*Supplementary file 1*).

To refine the candidate list of PCSK9-interacting proteins, we next analyzed cells expressing a fusion of BirA* with a control secreted protein, alpha-1 antitrypsin (A1AT). The interactome of A1AT showed substantial overlap with that of PCSK9 (108/162 proteins, *Figure 1C*). The A1AT cargo receptor LMAN1 was similarly labeled by both PCSK9-BirA* and A1AT-BirA*, suggesting that the restricted environment of the COPII vesicle may lead to nonspecific labeling of adjacent cargo receptors. We next compared the interactome of PCSK9 to that of SAR1A and SAR1B (*Figure 1D*), COPII proteins that localize to the cytoplasmic surface of budding COPII vesicles, identifying a total of 35 candidate ER cargo receptors interacting with both PCSK9 and either SAR1A or SAR1B (*Figure 1E–F*, *Supplementary file 1*). The majority of these candidates were annotated as integral membrane proteins (32/35, p=$3\times10^{-16}$) with localization in the ER (24/35, p=$1.6\times10^{-18}$), as would be expected for an ER cargo receptor (*Supplementary file 1*).

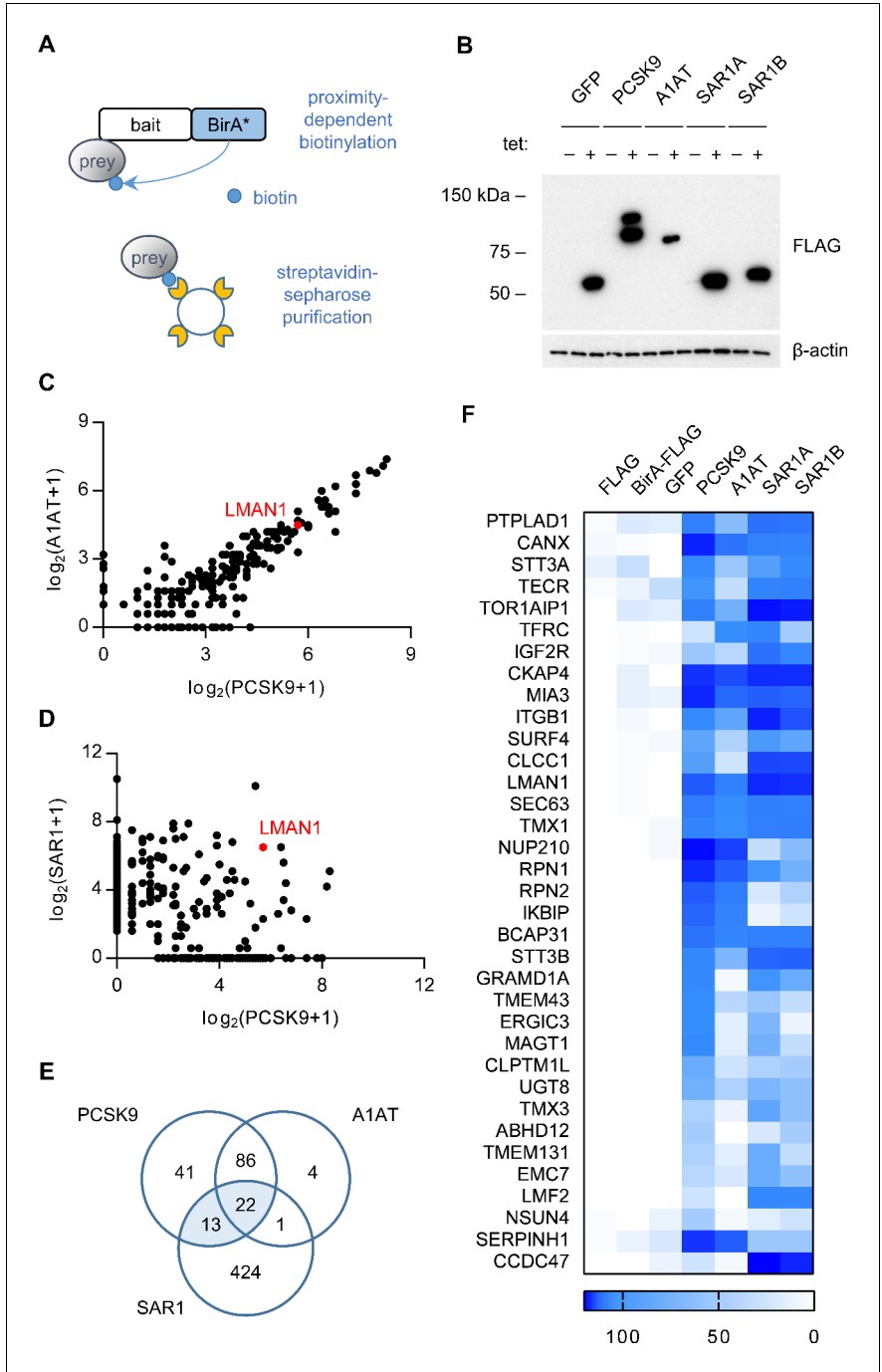

**Figure 1.** Proximity-dependent biotinylation with a PCSK9-BirA* fusion. (**A**) Proximity detection by mass spectrometry of streptavidin-purified prey proteins biotinylated by a fusion of BirA* to a bait protein of interest. (**B**) Immunoblotting of lysates of cells expressing various BirA*-fusion proteins. (**C**) Spectral counts of prey proteins identified from lysates of cells expressing PCSK9-BirA* relative to A1AT-BirA*. (**D**) Spectral counts of prey proteins purified from lysates of cells expressing PCSK9-BirA* relative to the maximum spectral count from lysates of cells expressing either SAR1A-BirA* or SAR1B-BirA*. (**E**) Venn diagram of identified prey proteins from lysates of cells expressing BirA* fusions with PCSK9, A1AT, or the maximum for either Sar1A or Sar1B. (**F**) Heat map of spectral counts for candidate proteins demonstrating interaction with both PCSK9-BirA* and either SAR1A-BirA* or SAR1B-BirA*. Spectral count values represent averages of 2 biologic replicates. Only prey proteins that exhibit BFDR ≤0.01 for one or more bait proteins are displayed. Source data is provided in *Supplementary file 1*.
DOI: https://doi.org/10.7554/eLife.38839.002

## A genome-scale CRISPR screen identifies SURF4 as a putative ER cargo receptor for PCSK9

We next developed a functional screen to identify genes involved in PCSK9 secretion (*Figure 2A*). We reasoned that mutants with reduced PCSK9 exit from the ER would accumulate intracellular PCSK9, and that fusion of PCSK9 to eGFP would allow for a cell-autonomous, scalable, and selectable readout of PCSK9 accumulation. We generated clonal HEK293T cell lines stably co-expressing both a PCSK9-eGFP fusion and, as an internal control, alpha-1 antitrypsin fused with mCherry. Immunoblotting verified the efficient secretion of both fusion proteins from clonal reporter cell lines (*Figure 2—figure supplement 1*).

Disruption of ER-Golgi transport by treatment of these reporter cells with brefeldin A, an Arf1 inhibitor, resulted in intracellular accumulation of both PCSK9-eGFP and A1AT-mCherry (*Figure 2B*). CRISPR-mediated inhibition of the ER cargo receptor for A1AT, LMAN1 (*Zhang et al., 2011*), resulted in intracellular accumulation of A1AT-mCherry with no effect on PCSK9-eGFP (*Figure 2B*). To screen for specific modifiers of PCSK9 secretion, we next sought to identify single guide RNAs (sgRNAs) that would induce accumulation of PCSK9-eGFP with no change in A1AT-mCherry

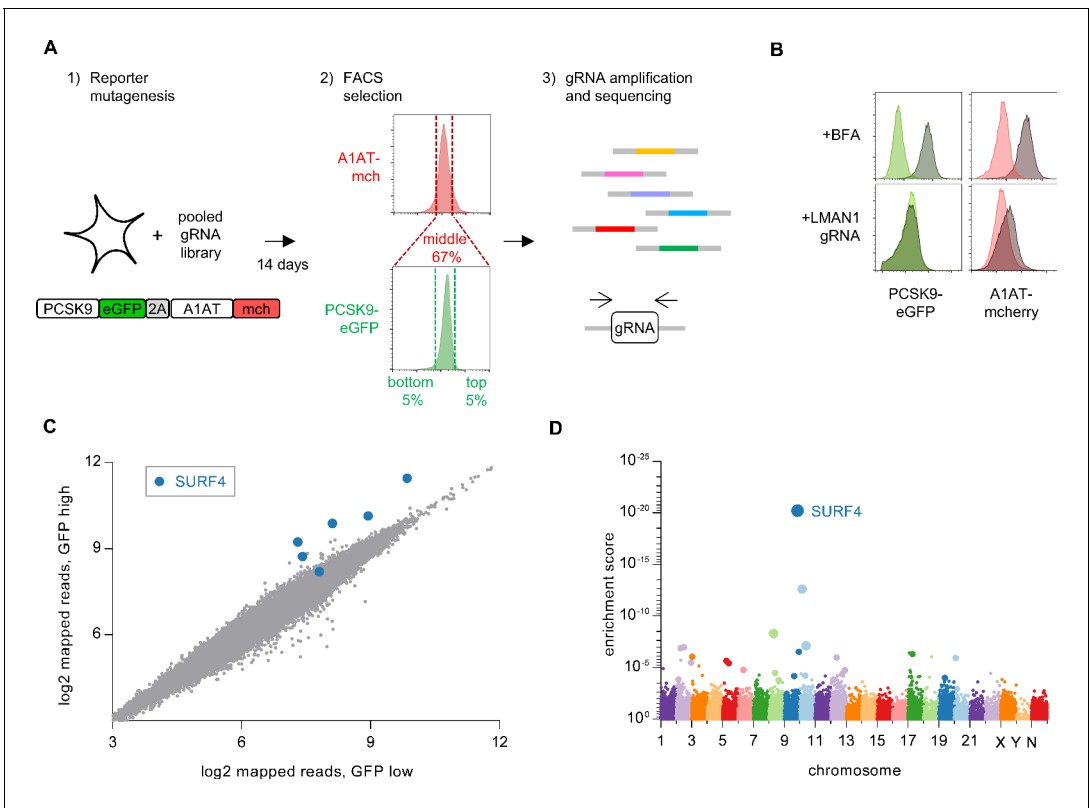

**Figure 2.** Whole genome CRISPR mutagenesis screen for PCSK9 secretion modifiers. (**A**) Strategy for whole genome screen. (**B**) Flow cytometry of reporter cells stably expressing PCSK9-eGFP-2A-A1AT-mCherry, treated with 1 μg/mL brefeldin A or a sgRNA targeting LMAN1. (**C**) Normalized abundance of each sgRNA in the library in eGFP high and eGFP low populations. (**D**) MAGeCK gene-level enrichment scores for each gene targeted by the library arranged by chromosome number and transcription start site. The diameter of the bubble is proportional to the number of unique sgRNAs targeting each gene that demonstrate significant enrichment in GFP high cells. Source data is provided in *Supplementary files 2* and *3*.

DOI: https://doi.org/10.7554/eLife.38839.003

The following figure supplements are available for figure 2:

**Figure supplement 1.** Analysis of PCSK9-eGFP-2A-A1AT-mCherry reporter cell clones.
DOI: https://doi.org/10.7554/eLife.38839.004

**Figure supplement 2.** Whole genome screen analysis.
DOI: https://doi.org/10.7554/eLife.38839.005

**Figure supplement 3.** Validation experiments for additional candidate genes.
DOI: https://doi.org/10.7554/eLife.38839.006

fluorescence. We mutagenized the PCSK9-eGFP-2A-A1AT-mCherry reporter cell line with the GeCKOv2 pooled library of 123,411 sgRNAs that includes six independent sgRNAs targeting nearly every coding gene in the human genome (*Sanjana et al., 2014*) (*Figure 2A*). Mutants with aberrant PCSK9-eGFP fluorescence but normal A1AT-mCherry fluorescence were then isolated by flow cytometry, with integrated lentiviral sgRNA sequences quantified by deep sequencing and analyzed for enrichment in PCSK9-eGFP high cells. The coverage and distribution of sgRNA sequencing reads demonstrated maintenance of library complexity and high reproducibility between biological replicates (*Figure 2—figure supplement 2*).

Strikingly, the four most enriched sgRNAs in the PCSK9-eGFP high cell population all targeted the same gene, *SURF4* (*Figure 2C*, *Supplementary file 2*). The enrichment of *SURF4*-targeting sgRNA in eGFP-high cells was consistent across each of 4 biologic replicates and, after adjustment for multiple hypothesis testing, statistically significant for 5 of the 6 *SURF4*-targeting sgRNAs in the library ($p < 10^{-13} – 10^{-36}$, *Figure 3A*). Gene-level analysis confirmed the strongest enrichment for *SURF4*-targeting sgRNA (*Figure 2D*, *Supplementary file 3*). SURF4 is a homologue of yeast Erv29, an ER cargo receptor that mediates the secretion of glycosylated pro-alpha-factor (*Belden and Barlowe, 2001*). Comparison of candidate PCSK9 cargo receptors identified by either CRISPR functional screening (*Figure 2D*) or proximity-dependent biotinylation (*Figure 1F*) demonstrated that SURF4 was the only candidate identified by both approaches.

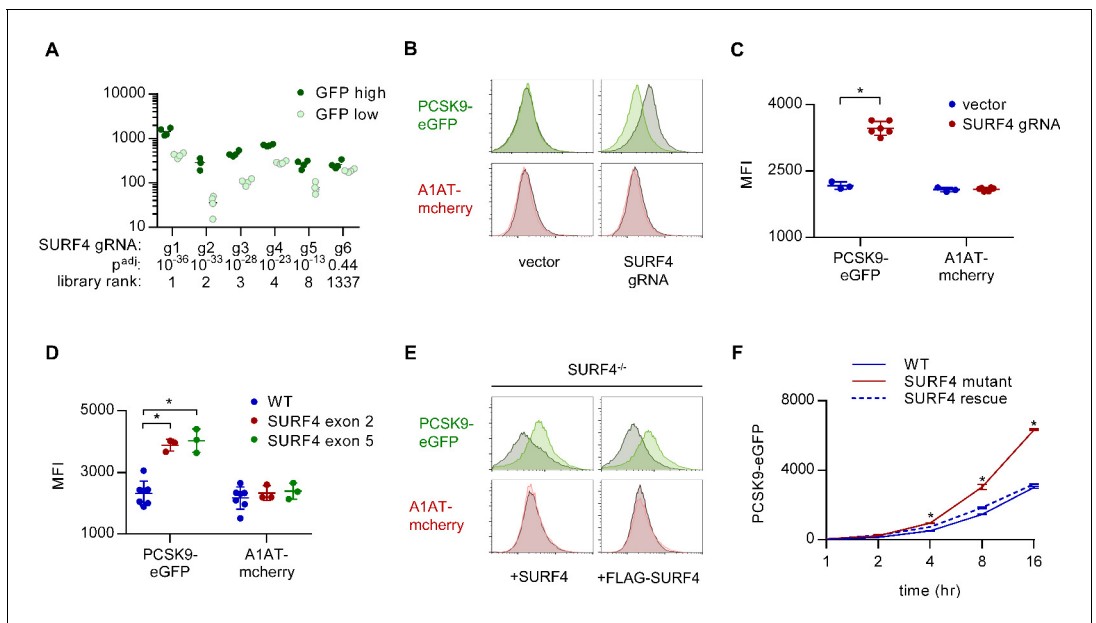

**Figure 3.** *SURF4* mutagenesis causes an accumulation of intracellular PCSK9-eGFP. (A) Individual sgRNA sequencing counts for *SURF4*-targeting sgRNA in eGFP high and eGFP low populations for each of 4 biologic replicates. Adjusted p values were calculated using DESeq2. (B) Flow cytometry histograms of PCSK9-eGFP and A1AT-mCherry fluorescence in reporter cells transfected with plasmids delivering Cas9 and *SURF4*-targeting sgRNA or empty vector. (C) Quantification of intracellular fluorescence for cells treated with empty vector (n = 3) or unique *SURF4*-targeting sgRNAs (n = 6). (D) Quantification of intracellular fluorescence for clonal cell lines each containing frameshift-causing indels at two different *SURF4* target sites (n = 7 wild-type clones, n = 3 clones generated from each *SURF4*-targeting sgRNA). (E) Flow cytometry histograms for cells expressing PCSK9-eGFP-2A-A1AT-mCherry and deleted for *SURF4* with or without stable expression of a wild-type or FLAG-tagged *SURF4* cDNA. (F) Time course of intracellular accumulation of tetracycline-inducible PCSK9-eGFP on WT, SURF4-deficient, or SURF4 rescue background (n = 3 biologic replicates for each cell line at each time point). *p<0.05 by Student's t-test. Error bars represent standard deviations.

DOI: https://doi.org/10.7554/eLife.38839.007

The following figure supplements are available for figure 3:

**Figure supplement 1.** ER stress markers.

DOI: https://doi.org/10.7554/eLife.38839.008

**Figure supplement 2.** SURF4-deficient genotypes.

DOI: https://doi.org/10.7554/eLife.38839.009

## SURF4 deletion causes intracellular accumulation of PCSK9-eGFP in HEK293T cells

To validate the functional interaction of SURF4 with PCSK9-eGFP, we generated plasmids encoding Cas9 and 6 independent *SURF4*-targeting sgRNAs (three from the original screen and three additional unique sgRNAs). Reporter cells were transiently transfected with each of these 6 *SURF4*-targeting constructs and analyzed by FACS, with all six sgRNAs resulting in accumulation of intracellular PCSK9-eGFP fluorescence with no effect on intracellular A1AT-mCherry fluorescence (*Figure 3B–C*) or induction of ER stress markers (*Figure 3—figure supplement 1*). Similarly, six clonal cell lines carrying sequence-verified indels in either *SURF4* exon 2 or exon 5 (*Figure 3—figure supplement 2*) all exhibited specific PCSK9-eGFP accumulation with no effect on A1AT-mCherry (*Figure 3D*). This phenotype was rescued by stable expression of wild-type *SURF4* cDNA (*Figure 3E*). The intracellular accumulation of PCSK9 upon *SURF4* disruption was confirmed in an independently-derived HEK293T cell line carrying an inducible PCSK9-eGFP allele. Accumulation of PCSK9-eGFP in *SURF4* mutant cells relative to wild-type cells was detectable within 4 hr of induction (*Figure 3F*).

To identify other potential cellular components also required for efficient PCSK9 secretion, we next examined the 21 genes in addition to *SURF4* which exhibited potentially significant enrichment scores (FDR $\leq$ 10%) in the whole genome CRISPR screen. PCSK9-eGFP-2A-A1AT-mCherry reporter cells were transduced with individual lentiviral CRISPR constructs targeting each of these genes and analyzed by FACS. None of the sgRNA targeting these additional genes were found to result in a significant effect specifically on PCSK9-eGFP fluorescence (*Figure 2—figure supplement 3*). Thus, within the statistical power of our screen, SURF4 emerges as the single gene out of the ~19,000 human genes targeted by the GeCKOv2 library (*Sanjana et al., 2014*) whose inactivation results in specific intracellular retention of PCSK9 in HEK293T cells.

## SURF4 localizes to the early secretory pathway where it physically interacts with PCSK9

HEK293T cells were engineered to stably express SURF4 with an N-terminal FLAG epitope tag. FLAG-tagged SURF4 demonstrated similar rescue of PCSK9-eGFP fluorescence in SURF4-deficient cells compared to native, untagged SURF4 (*Figure 3E*), demonstrating that this tag does not interfere with SURF4 function. Consistent with previous reports (*Mitrovic et al., 2008*; *Saegusa et al., 2018*) and compatible with a role for SURF4 as an ER cargo receptor, immunofluorescence of FLAG-SURF4 demonstrated colocalization with a marker of the ER and, to a lesser extent, the ERGIC compartment (*Figure 4A–B*). FLAG-tagged SURF4 and GFP-tagged PCSK9 were found to co-immunoprecipitate from cell lysates prepared in the presence of the chemical crosslinker dithiobis (succinimidyl propionate), with no detectable nonspecific co-immunoprecipitation for several control ER and ERGIC-localized proteins (*Figure 4D*).

## Loss of SURF4 results in decreased PCSK9 secretion and ER accumulation of PCSK9

Fluorescence assays on both extracellular conditioned media and intracellular lysates prepared from both wild-type and SURF4-deficient cells demonstrated a significantly decreased ratio of extracellular to intracellular PCSK9-eGFP fluorescence (*Figure 5A*), consistent with a defect in extracellular secretion. Pulse-chase labeling confirmed a decreased rate of PCSK9-eGFP secretion into the conditioned media in SURF4-deficient cells (*Figure 5—figure supplement 1*).

To exclude interaction of SURF4 with the GFP portion of the fusion rather than PCSK9 itself, we also examined SURF4-dependence for the secretion of untagged PCSK9. Stable cell lines were generated by Flp/FRT-mediated knock-in of PCSK9 coding sequence into a tetracycline-inducible locus of HEK293T cells that were either wild-type or SURF4-deficient, the latter with or without stable integration of a wild-type *SURF4* cDNA expression construct. Consistent with the reduced secretion observed above for the PCSK9-eGFP fusion, untagged PCSK9 also exhibited a significant decrease in the ratio of extracellular to intracellular levels in SURF4-deficient cells that was rescued by expression of a *SURF4* transgene (*Figure 5B–C*). Quantitative PCR demonstrated equivalent levels of PCSK9 mRNA levels in wild-type and SURF4-deficient cells (*Figure 5D*), excluding an indirect effect on PCSK9 transcription or mRNA stability.

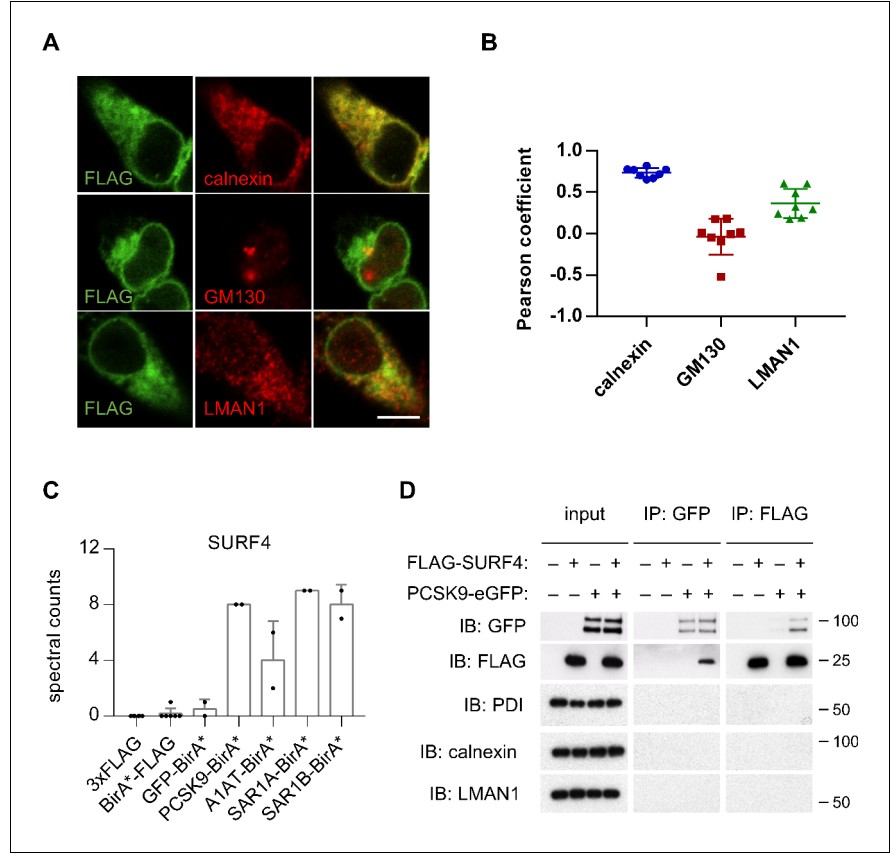

**Figure 4.** SURF4 localizes to the early secretory pathway where it physically interacts with PCSK9. (**A**) Immunofluorescence of FLAG-SURF4 together with markers of the ER (calnexin), ERGIC (LMAN1), and Golgi (GM130). Scale bar = 10 μm. (**B**) Quantification of colocalization (n = 8 cells analyzed for each combination of antibody staining). (**C**) Spectral counts for SURF4 in streptavidin-purified eluates from cells expressing various BirA* fusion proteins. (**D**) Immunoprecipitations were performed using antibodies directed against FLAG or GFP from lysates of cells expressing FLAG-SURF4, PCSK9-eGFP, both, or neither. Error bars represent standard deviations. DOI: https://doi.org/10.7554/eLife.38839.010

To characterize the compartmental localization of intracellular PCSK9, we performed live cell fluorescence microscopy on wild-type and SURF4-deficient cells. PCSK9-eGFP fluorescence demonstrated increased colocalization with an ER marker in SURF4-deficient cells, consistent with ER retention (*Figure 5E–F*). Similarly, the predominant form of native PCSK9 in SURF4-deficient cells was sensitive to endoglycosidase H (*Figure 5G–H*, *Figure 5—figure supplement 2*), consistent with its localization in the ER (*Freeze and Kranz, 2010*). The relative intensity of endoH-resistant PCSK9 was unchanged, consistent with continued normal post-Golgi transport and secretion. Collectively these results indicate that SURF4 promotes the efficient ER exit and secretion of PCSK9.

## Discussion

It is estimated that ~3000 human proteins are extracellularly secreted through the COPII pathway (*Uhlén et al., 2015*), though few ER cargo receptors for these proteins have been identified and the proportion of secreted proteins that depend upon cargo receptor interactions is unknown (*Barlowe and Helenius, 2016*). Our findings suggest that SURF4 actively recruits PCSK9 into COPII vesicles, providing additional support for the cargo capture model of protein secretion. *SURF4*-deleted cells exhibit only a partial defect in PCSK9 secretion and whether the residual *SURF4*-independent secretion is due to bulk flow, or interactions with alternative ER cargo receptors, remains unknown.

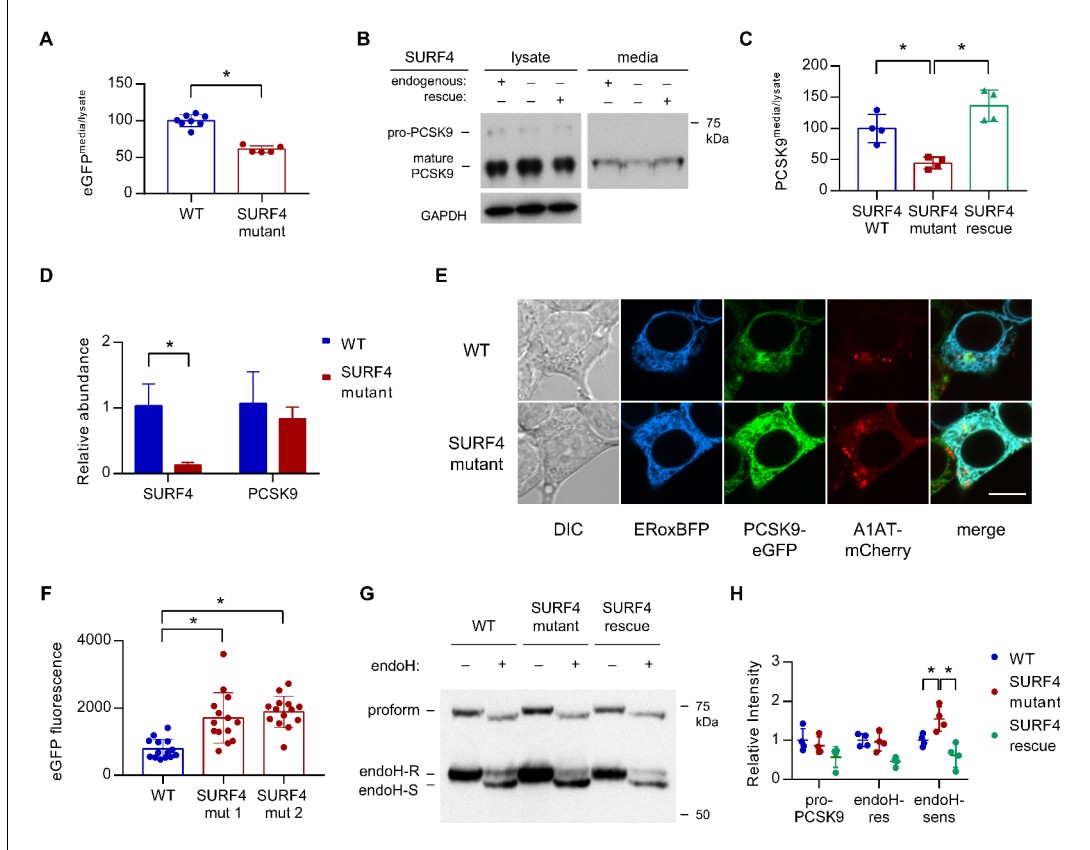

**Figure 5.** *SURF4* mutagenesis causes a decrease in PCSK9 extracellular secretion and an accumulation of PCSK9 in the ER. (A) Fluorescence detection of PCSK9-eGFP in extracellular conditioned media relative to cellular lysate in WT (n = 7) and clonal SURF4-deficient (n = 5) fluorescent reporter cell lines. (B) Immunoblotting of tetracycline-inducible PCSK9 in extracellular conditioned media and cellular lysates from WT, SURF4-deficient, or SURF4 rescued cells. (C) Quantification of densitometry of native PCSK9 relative in conditioned media and cellular lysates, normalized to GAPDH, for WT, SURF4-deficient, or SURF4 rescued cells (n = 4 biologic replicates for each cell line). (D) Quantitative PCR of *SURF4* and *PCSK9* transcript levels from RNA isolated from a *SURF4* WT or mutant fluorescent reporter cell line (n = 3 measurement replicates). (E) Live cell fluorescence microscopy of fluorescent reporter cells, either WT or SURF4-deficient, transfected with the ER marker ERoxBFP. Scale bar = 10 μm. (F) Quantification of PCSK9-eGFP signal intensity in pixels positive for ERoxBFP fluorescence (n = 14 for each cell line). (G) EndoH-sensitivity of PCSK9 expressed in WT, SURF4-deficient, or SURF4 rescued cells. (H) Quantification of endoH-sensitivity, normalized to average intensity of average WT band intensity (n = 4 biologic replicates for each cell line). *p<0.05 by Student's t-test. Error bars represent standard deviations.

DOI: https://doi.org/10.7554/eLife.38839.011

The following figure supplements are available for figure 5:

**Figure supplement 1.** Pulse-chase labeling of PCSK9-eGFP secretion.

DOI: https://doi.org/10.7554/eLife.38839.012

**Figure supplement 2.** EndoH-sensitivity of PCSK9 expressed in WT, SURF4-deficient, or SURF4 rescued cells for each of 4 independent biologic replicates.

DOI: https://doi.org/10.7554/eLife.38839.013

To potentially circumvent the expected transient and/or low affinity nature of interactions between cargoes and their receptors, we applied purification by BirA*-mediated proximity-dependent biotinylation, which converts transient protein-protein interactions into permanent modifications (*Roux et al., 2012*). Although this approach led to the detection of SURF4 interactions with PCSK9, SAR1A, and SAR1B, SURF4 was similarly marked by A1AT. Likewise, the A1AT cargo receptor LMAN1 was marked by both A1AT and PCSK9. These data suggest that BirA*-mediated proximity-dependent biotinylation may efficiently label ER cargo receptors, but lack the spatiotemporal resolution to distinguish incorporation within the same COPII vesicle from specific cargo – cargo receptor interactions.

CRISPR/Cas9-mediated gene editing has emerged as an efficient, programmable, and high-throughput tool for forward genetic screening in mammalian cells (*Shalem et al., 2015*). Our discovery of the PCSK9-SURF4 interaction demonstrates the feasibility of identifying ER cargo receptors by functional genomic screening and suggests that pooled CRISPR screening may prove to be a generalizable strategy applicable to the thousands of other secreted proteins for which a cargo receptor has not yet been identified. A limitation of our approach is its reliance on an immortalized cell line with the potential for accumulated somatic genetic and epigenetic changes, as well as restriction to a single cell-type gene expression program. Though LMAN1 cargoes appear to retain their cargo receptor-dependence for secretion in a variety of distantly related cell types (*Nyfeler et al., 2008*; *Zhang et al., 2005*; *Nyfeler et al., 2006*), the same many not apply to SURF4.

Of note, SEC24A and/or SEC24B were previously shown to be required for efficient PCSK9 secretion in mice (*Chen et al., 2013*). Though these genes were not identified in the current screen, this observation could be explained by the 5 – 10 fold greater expression of SEC24A than SEC24B in mouse liver, in contrast to the similar expression of both paralogs in HEK293T cells (*Sultan et al., 2008*).

In addition to a cargo receptor facilitating ER exit, our screening strategy should also survey other segments of the secretory pathway. Though sortilin has previously been implicated in post-Golgi transport of PCSK9 (*Gustafsen et al., 2014*), sortilin dependence was not confirmed in a subsequent study (*Butkinaree et al., 2015*) and also was not detected in the current screen. However, our data do not exclude a broader role for sortilin that also affected A1AT secretion, consistent with its contribution to the secretion of other proteins, including gamma-interferon (*Herda et al., 2012*) and ApoB-100 (*Kjolby et al., 2010*).

The yeast homologue of SURF4, Erv29p, was originally identified by proteomic analysis of purified COPII vesicles (*Otte et al., 2001*). Erv29p promotes the secretion and concentrative sorting of the yeast mating factor gpαf into COPII vesicles through interaction with a hydrophobic I-L-V signal in gpαf (*Belden and Barlowe, 2001*; *Malkus et al., 2002*; *Otte and Barlowe, 2004*). In HeLa cells, SURF4 has been shown to cycle in the early secretory pathway and to interact with LMAN1 and members of the p24 family. RNAi-mediated knockdown of SURF4 alone resulted in no overt phenotype, though when combined with knockdown of LMAN1 caused morphologic changes to the ERGIC and Golgi compartments (*Mitrovic et al., 2008*). SURF4 has also been shown to interact with STIM1 and modulate store-operated calcium entry, though the mechanism underlying this observation is unclear (*Fujii et al., 2012*). Our screen did not identify significant enrichment of sgRNAs targeting p24 proteins, LMAN1, or STIM1, suggesting that these interactions are not required for efficient PCSK9 secretion in HEK293T cells.

Taken together with the marked reductions in plasma cholesterol associated with genetic or therapy-induced reductions in plasma PCSK9 (*ODYSSEY LONG TERM Investigators et al., 2015*), our findings raise the possibility that SURF4 could represent an additional novel therapeutic target for the treatment of hypercholesterolemia. However, *SURF4* has not been identified in genome-wide association studies for human lipid phenotypes (*Global Lipids Genetics Consortium et al., 2013*; *Lange et al., 2015*), suggesting that partial reduction of SURF4 expression does not limit PCSK9 secretion, consistent with the normal PCSK9 secretion and cholesterol profiles of *Sec24A*$^{+/-}$ mice (*Chen et al., 2013*) and the normal levels of LMAN1 cargoes (coagulation factors V, VIII, and A1AT) in LMAN1$^{+/-}$ mice (*Zhang et al., 2011*). Though a loss-of-function variant (p.Gln185Ter) is present in ~1:500 individuals (*Exome Aggregation Consortium et al., 2016*), no human diseases have been associated with SURF4 deficiency and no mouse models for *Surf4* deletion have been reported (*Online Mendelian Inheritance in Man, OMIM, 2018*). Erv29, the SURF4 homolog in yeast, is required for gpαf secretion (*Belden and Barlowe, 2001*), with a recent report demonstrating a role for the *C. elegans* homolog (SFT-4) in facilitating the secretion of yolk lipoproteins, as well as mammalian SURF4 in mediating apolipoprotein B secretion in HepG2 cells (*Saegusa et al., 2018*). The latter result, together with our findings, suggest a potentially broader role for SURF4 in the complex regulation of mammalian lipid homeostasis in vivo.

# Materials and methods

## Key resources table

| Reagent type (species) or resource | Designation | Source or reference | Identifiers | Additional information |
|---|---|---|---|---|
| Gene (*Homo sapiens*) | PCSK9 | NA | Uniprot Q8NBP7 | |
| Gene (*Homo sapiens*) | A1AT | NA | Uniprot P01009 | |
| Gene (*Homo sapiens*) | SURF4 | NA | Uniprot O15260 | |
| Cell line (*Homo sapiens*) | HEK293T | ATCC | RRID:CVCL_1926 | |
| Cell line (*Homo sapiens*) | T-Rex-293 | Invitrogen | RRID:CVCL_D585 | |
| Antibody | anti-PCSK9 (rabbit polyclonal) | Cayman Chemical | RRID:AB_569536 | (1:1000) |
| Antibody | anti-GFP (rabbit monoclonal) | Abcam | RRID:AB_303395 | (1:5000) |
| Antibody | anti-mCherry (rabbit polyclonal) | Abcam | RRID:AB_2571870 | (1:1000) |
| Antibody | anti-GAPDH (rabbit monoclonal) | Abcam | RRID:AB_2630358 | (1:10,000) |
| Antibody | anti-β-actin (mouse monoclonal) | Santa Cruz | RRID:AB_2714189 | (1:10,000) |
| Antibody | anti-LMAN1 (rabbit monoclonal) | Abcam | RRID:AB_10973984 | (1:5000) |
| Antibody | anti-calnexin (rabbit monoclonal) | Cell Signaling | RRID:AB_2228381 | (1:2000) |
| Antibody | anti-GM130 (rabbit monoclonal) | Abcam | RRID:AB_880266 | (1:1000) |
| Antibody | anti-BiP (rabbit monoclonal) | Abcam | RRID:AB_10859806 | (1:5000) |
| Antibody | anti-PDI (rabbit monoclonal) | Cell Signaling | RRID:AB_2156433 | (1:1000) |
| Antibody | HRP-conjugated anti-FLAG (goat polyclonal) | Abcam | RRID:AB_299061 | (1:10,000) |
| Antibody | HRP-conjugated anti-mouse secondary (goat polyclonal) | BioRad | RRID:AB_11125547 | (1:5000) |
| Antibody | HRP-conjugated anti-rabbit IgG (goat polyclonal) | BioRad | RRID:AB_11125142 | (1:5000) |
| Antibody | FITC-conjugated anti-FLAG (mouse monoclonal) | Sigma | RRID:AB_439701 | (1:500) |
| Antibody | Alexa647-conjugated anti-rabbit secondary (donkey polyclonal) | Abcam | ab150075 | (1:500) |
| Recombinant DNA reagent | pLentiCRISPRv2 | Addgene | 52961 | |
| Recombinant DNA reagent | pX459 | Addgene | 62988 | |
| Recombinant DNA reagent | pNLF-C1 | Promega | E1361 | |
| Recombinant DNA reagent | pDEST-pcDNA5-BirA-FLAG-C-term | PMID 24255178 | | |
| Recombinant DNA reagent | pENTR/D-TOPO | Invitrogen | K2400-20 | |

*Continued on next page*

Continued

| Reagent type (species) or resource | Designation | Source or reference | Identifiers | Additional information |
|---|---|---|---|---|
| Chemical compound, drug | brefeldin A | BioLegend | 420601 | |
| Chemical compound, drug | dithiobis (succinimidyl propionate) | Pierce | 22586 | |

## Cells and reagents

HEK293T cells were purchased from ATCC (Manassas VA). T-REx-293 cells were purchased from Invitrogen. Cell lines were validated by the AMPFLSTR Identifiler Plus Assay (Applied Biosystems, Foster City CA) and tested by the MycoAlert Mycoplasma Detection Kit (Lonza, Basel Switzerland). Cells were cultured in DMEM (Invitrogen, Carlsbad CA) containing 10% FBS (D10) in a humidified 37°C chamber with 5% CO2. The expression construct for PCSK9-eGFP-2A-A1AT-mCherry was generated by Gibson assembly (*Gibson et al., 2009*) of vector sequence derived from pNLF-C1 (Promega, Madison WI) and PCSK9 and A1AT cDNA derived by RT-PCR from HepG2 mRNA. Expression constructs for PCSK9, A1AT, SAR1A, and SAR1B fused to BirA* were generated by cDNA ligation into the entry vector pENTR/D-TOPO (Invitrogen) and Gateway cloning into the destination vector pDEST-pcDNA5-BirA-FLAG C-term (*Couzens et al., 2013*) using LR clonase II (Invitrogen). This vector was also used as a backbone for the Gibson assembly of tetracycline-inducible expression of native PCSK9 and PCSK9-eGFP. For CRISPR experiments, sgRNA sequences were ligated into pLentiCRISPRv2 (Addgene #52961, a gift from Feng Zhang (*Sanjana et al., 2014*)) or pX459 (Addgene #62988, a gift from Feng Zhang) using BsmBI or BbsI restriction enzyme sites, respectively. Transfections were performed with FugeneHD (Promega) or Lipofectamine 3000 (Invitrogen) per manufacturer's instructions. Where indicated, clonal cell lines were derived by diluting cell suspensions to a single cell per well and expanding individual wells. Genotyping of clonal cell lines was performed by Sanger sequencing of target site PCR amplicons of genomic DNA isolated by QuickExtract (Epicentre, Madison WI). The pLentiCRISPRv2 whole genome CRISPR library (Addgene #1000000048, a gift from Feng Zhang [*Sanjana et al., 2014*]) was expanded by eight separate electroporations for each half library into Endura electrocompetent cells (Lucigen, Middleton WI), plated on 24.5 cm bioassay plates, and pooled plasmids isolated by HiSpeed Maxi Prep (Qiagen, Hilden Germany). The pooled lentiviral library was prepared by co-transfecting 120 μg of each half library together with 120 μg of pCMV-VSV-G (Addgene #8454, a gift from Bob Weinberg [*Stewart et al., 2003*]) and 180 μg psPAX2 (Addgene #12260, a gift from Didier Trono) into a total of 12 T225 tissue culture flasks of ~70% confluent HEK293T cells using FugeneHD per manufacturer's instructions. Media was replaced at 12 hr post-transfection with D10 supplemented with 1% BSA, which was collected and changed at 24, 36, and 48 hr. Harvested media was centrifuged at 1000 g for 10 min, pooled and filtered through a 0.45 μm filter, aliquoted, snap-frozen with liquid nitrogen and stored at −80°C until the time of use.

## Whole genome CRISPR screen

For each of 4 independent biological replicates, a total of ~90 million cells stably expressing PCSK9-eGFP-2A-A1AT-mCherry were transduced at a multiplicity of infection of ~0.3 with the whole genome CRISPR library. Puromycin selection (1 μg/mL) was applied from day 1 to day four post-transduction. Cells were passaged every 2 – 3 days and maintained in logarithmic phase of growth. After 14 days, a total of ~240 million cells were detached with TrypLE Express (Invitrogen), harvested in D10 at 4°C, collected by centrifugation at 500 g for 5 min, the pellet resuspended in 4°C phosphate-buffered saline (PBS) and filtered through a 35 μm nylon mesh into flow cytometry tubes, which were kept on ice until the time of sorting. A BD FacsAriaII was used to isolate cell populations containing ~7 million cells per subpopulation into tubes containing D10. Genomic DNA was isolated using a DNEasy purification kit (Qiagen). Integrated lentiviral sgRNA sequences were then amplified using Herculase II polymerase (Agilent, Santa Clara CA) in a two step PCR reaction as previously described (*Sanjana et al., 2014*; *Shalem et al., 2014*) with 20 cycles for round 1 PCR and 14 cycles for round 2. Amplicons were then sequenced on a HiSeq Rapid Run (Illumina, San Diego CA), with 95.2% of clusters passing quality filtering to generate a total of ~210 million reads with a mean

quality score of 34.99. Individual sgRNA sequences were mapped with a custom Perl script (*Source code 1*) that seeded sequences onto a constant 24 nucleotide region upstream of the variable 20mer sgRNA, allowing up to one nucleotide mismatch, and reading the upstream six nucleotide barcode and downstream 20 nucleotide sgRNA sequence, which was then mapped to the library reference database with no mismatches tolerated. Enrichment was assessed using DESeq2 for individual sgRNA sequences (*Love et al., 2014*) and MAGeCK for gene-level computations (*Li et al., 2014*).

## PCSK9 secretion assays

Clonal cell lines either wild-type for *SURF4* or containing frameshift-causing *SURF4* indels were isolated on a background of the HEK293T fluorescent reporter cell line or in T-REx-293 cells with a Flp/FRT-integrated PCSK9 cDNA. Cells were seeded at equal density in 10 cm plates and cultured in D10 for non-fluorescence-based assays or Fluorobrite media (ThermoFisher, Waltham MA) supplemented with 10% FBS for fluorescence-based assays. At the time of analysis, conditioned media was removed, clarified by centrifugation for 10 min at 1000 g, and supernatant analyzed immediately or stored at −20°C. Cell monolayers were detached with trypLE express, collected in fresh media, pelleted, washed with PBS, and resuspended in 750 μL RIPA buffer (ThermoFisher) supplemented with protease inhibitors (Roche, Basel Switzerland). RIPA lysates were briefly sonicated, rotated end-over-end for 45 min at 4°C, and centrifuged at ~20,000 g for 30 min. Supernatants were transferred to a new tube and analyzed immediately or stored at −20°C. For fluorescence assays, samples were measured in triplicate in 96 well plates using an EnSpire fluorescence plate reader (PerkinElmer, Waltham MA), with fluorescent intensity zeroed on the autofluorescence of parental cells not expressing fluorescent fusion proteins for lysates or unconditioned media for conditioned media. For immunoblotting, samples were probed with antibodies against PCSK9 (Cayman 10007185, 1:1000), GAPDH (Abcam, Cambridge UK, ab181602, 1:10,000), GFP (Abcam, ab290, 1:5000), β-actin (Santa Cruz, sc-47778, 1:10,000), FLAG (Abcam, ab1238, 1:10,000), mCherry (Abcam, ab167453, 1:1000), LMAN1 (Abcam, ab125006, 1:5000), calnexin (Cell Signaling, 2679, 1:2000), BiP (Abcam, ab108613, 1:5000), PDI (Cell Signaling, 3501, 1:1000). Densitometry was performed with ImageJ software (*Schneider et al., 2012*).

## Endoglycosidase H assays

To test for the N-glycosylation state of PCSK9, approximately 100 μg of RIPA lysate was incubated with denaturation buffer (NEB, Ipswich MA) for 10 min at 95°C, then split in half and incubated with or without 0.5 μL of PNGase (NEB) or EndoH (NEB) for 1 hr at 37°C. Laemmli sample buffer (*Laemmli, 1970*) was added, samples boiled for 5 min, resolved on a 10% Tris-HCl polyacrylamide gel, and analyzed by immunoblotting as above.

## Microscopy

Cells were grown on 35 mm poly-D lysine-coated glass bottom dishes (MatTek, Ashland MA). For live cell microscopy, cells were transiently transfected with an expression plasmid for ERoxBFP (*Costantini et al., 2015*) (Addgene #68126, a gift from Erik Snapp (*Costantini et al., 2015*)) and visualized at 24 – 48 hr post-transfection. For immunostaining, cells were washed with PBS, fixed with 2% paraformaldehyde, permeabilized with 0.1% Triton X-100 in PBS, blocked with 1% BSA and 0.1% Tween-20 in PBS, stained with FITC-conjugated anti-FLAG antibody (Sigma, St. Louis MO, F4049) and unconjugated rabbit antibodies against either calnexin (Cell Signaling Technology, Danvers MA, #2679), LMAN1 (Abcam, ab125006), or GM130 (Abcam, ab52649), then stained with Alexa647-conjugated anti-rabbit secondary antibody (Abcam, ab150075). All fluorescent imaging was performed on a Nikon A2 confocal microscope. Colocalization quantification was performed with Nikon Elements software. For all microscopy experiments, the observer was blinded to cell genotype and only unblinded after completion of quantitative analysis.

## BioID and mass spectrometry

BioID and mass spectrometry analysis was performed essentially as described (*Chapat et al., 2017*). Briefly, stable HEK293 Flp-In T-REx cells were grown on 15 cm plates to approximately 75% confluence. Bait expression and proximity labeling were induced by the addition of tetracycline (1 μg/mL)

and biotin (50 µM) and proceeded for 24 hr. Cells were collected in PBS and biotinylated proteins purified by streptavidin-agarose affinity purification. Proteins were digested on-bead with sequencing-grade trypsin in 50 mM ammonium bicarbonate pH 8.5. Peptides were acidified by the addition of formic acid (2% (v/v) final) and dried by vacuum centrifugation. Dried peptides were suspended in 5% (v/v) formic acid and analysed on a TripleTOF 6600 mass spectrometer (SCIEX) in-line with a nanoflow electrospray ion source and nano-HPLC system. Raw data were searched and analyzed within ProHits LIMS (*Liu et al., 2010*) and peptides matched to genes to determine prey spectral counts (*Liu et al., 2016*). High confidence proximity interactions (BFDR ≤1%) were determined through SAINT analysis (*Teo et al., 2014*) implemented within ProHits. Bait samples (biological duplicates) were compared against 14 independent negative control samples (2 BirA*-FLAG-GFP only, 6 BirA*-FLAG only, and 6 3xFLAG only expressing cell lines) which were 'compressed' to six virtual controls to increase the stringency in scoring (*Mellacheruvu et al., 2013*). Data has been deposited as a complete submission to the MassIVE repository (https://massive.ucsd.edu/ProteoSAFe/static/massive.jsp) and assigned the accession number MSV000082222. The ProteomeXchange accession is PXD009368.

## Mass spectrometry data analysis

All raw (WIFF and WIFF.SCAN) files were saved in our local interaction proteomics LIMS, ProHits (*Liu et al., 2010*). mzXML files were generated from raw files using the ProteoWizard (v3.0.4468) and SCIEX converter (v1.3 beta) converters, implemented within ProHits. The searched database contained the human complement of the RefSeq protein database (version 57) complemented with SV40 large T-antigen sequence, protein tags, and common contaminants (72,226 sequences searched including decoy sequences). mzXML files were searched by Mascot (v2.3.02) and Comet (v2016.01 rev. 2) with up to two missed trypsin cleavage sites allowed and methionine oxidation and asparagine/glutamine deamidation set as variable modifications. The fragment mass tolerance was 0.15 Da and the mass window for the precursor was ±30 ppm with charges of 2 + to 4+ (both mono-isotopic mass). Search engine results were analyzed using the Trans-Proteomic Pipeline (TPP v4.6 OCCUPY rev three check) (*Deutsch et al., 2010*) via iProphet (*Shteynberg et al., 2011*). Peptides with PeptideProphet scores ≥ 0.85 were mapped back to genes (gene IDs were from RefSeq). If peptides were shared between multiple genes, spectral counts were assigned exclusively to those genes with unique peptide assignments proportionally to the evidence for that assignment. If peptides matched only to genes without unique peptide assignments, spectral counts were divided equally between those genes (*Liu et al., 2016*). SAINTexpress (v3.6.1) (*Teo et al., 2014*) was used to calculate the probability that identified proteins were significantly enriched above background contaminants using spectral counting (semi-supervised clustering) through comparing bait runs to negative control runs.

## Immunoprecipitation

Cells were harvested and resuspended at a density of ~$5 \times 10^6$ cells/mL in PBS supplemented with 2 mM CaCl$_2$ and 2 mM dithiobis(succinimidyl propionate) (Pierce, Waltham MA), rotated end-over-end at 4°C, quenched with the addition of Tris-HCl (pH 7.5) to a final concentration of 25 mM, pelleted, and resuspended in IP Lysis Buffer (50 mM Tris-HCl, 150 mM NaCl, 2 mM CaCl$_2$, and 1.0% Triton X-100, supplemented with protease inhibitors (Roche), pH 7.5). Lysates were prepared as described above. Immunoprecipitation from lysates was performed with M2-FLAG affinity gel (Sigma) or GFP-trap magnetic beads (Chromo-Tek, Hauppage NY). To reduce nonspecific binding, M2-FLAG affinity gel was pre-blocked with 1 hr incubation in IP Blocking Buffer (50 mM Tris-HCl, 500 mM NaCl, 2 mM CaCl$_2$, 5% BSA, pH 7.5). Pulldowns were performed with 500 µL of lysate with end-over-end rotation at 4°C overnight. A total of 5 washes were performed with IP Lysis Buffer for GFP-trap beads or IP Blocking Buffer for M2-FLAG affinity gel. Protein elution was performed by incubating beads at room temperature for 15 min with 2X Laemmli sample buffer supplemented with β–mercaptoethanol.

## Screen validation

All genes identified by MAGeCK with a FDR < 10% were selected for follow up validation. The top-ranking sgRNA for each gene was individually cloned into BsmBI sites of pLentiCRISPRv2. Individual

lentiviral stocks were prepared and used to transduce fluorescent reporter cells at an MOI <0.5, followed by puromycin selection, and passaging for 2 weeks prior to FACS analysis. The mean fluorescence intensity of a total of 20,000 gated events was recorded for each construct and compared to the mean intensity of 3 nontargeting sgRNA constructs. A total of 3 biologic replicates was performed.

## Pulse-chase analysis

Cells were seeded into six well plates and induced with 1 μg/mL tetracycline (included in each of the media preparations below) overnight before near-confluent monolayers were washed and incubated with Starvation Media (DMEM lacking cysteine and methionine (Invitrogen) and supplemented with tetracycline with 10% dialyzed FBS (Fisher)) for 20 min at 37°C. Starvation Media was then replaced with Pulse Media (Starvation Media supplemented with 75 μCi/sample EXPRE$^{35}$S$^{35}$S Protein Labeling Mix (PerkinElmer)) and cells were incubated for 30 min at 37°C. Cells were then washed and incubated with Chase Media (Starvation Media supplemented with 5 mM each of unlabeled methionine and cysteine) for the indicated time points, after which conditioned media (2 mL per sample) and cellular lysates (collected in 1 mL lysis buffer) were prepared as described above for co-immunoprecipitation experiments. For each immunoprecipitation, 20 μL of GFP-trap beads were used with either 200 μL of cellular lysate of 400 μL of conditioned media. Proteins were eluted in 50 μL of sample buffer, of which 10 μL was analyzed by SDS-PAGE and autoradiography.

## Statistical analysis

The statistical significance of differences in quantitative data between control and experimental groups was calculated using the Student's $t$-test. CRISPR screen and mass spectrometry data analysis was performed as described above.

## Acknowledgments

This research was supported by NIH grants R35-HL135793T (DG) and T32-HL007853 (BTE). D G is a Howard Hughes Medical Institute investigator.

## Additional information

### Competing interests

David Ginsburg: Reviewing editor, *eLife*. The other authors declare that no competing interests exist.

### Funding

| Funder | Grant reference number | Author |
|---|---|---|
| National Heart, Lung, and Blood Institute | R35-HL135793T | David Ginsburg |
| National Heart, Lung, and Blood Institute | T32-HL007853 | Brian Emmer |

The funders had no role in study design, data collection and interpretation, or the decision to submit the work for publication.

### Author contributions

Brian T Emmer, Conceptualization, Data curation, Formal analysis, Investigation, Methodology, Writing—original draft, Writing—review and editing; Geoffrey G Hesketh, Data curation, Formal analysis; Emilee Kotnik, Vi T Tang, Paul J Lascuna, Jie Xiang, Data curation; Anne-Claude Gingras, David Ginsburg, Conceptualization, Formal analysis, Writing—review and editing; Xiao-Wei Chen, Formal analysis

## Author ORCIDs

Brian T Emmer (iD) http://orcid.org/0000-0001-7365-1021
Vi T Tang (iD) http://orcid.org/0000-0001-6079-9756
Anne-Claude Gingras (iD) http://orcid.org/0000-0002-6090-4437
Xiao-Wei Chen (iD) http://orcid.org/0000-0003-4564-5120
David Ginsburg (iD) http://orcid.org/0000-0002-6436-8942

## Decision letter and Author response

Decision letter https://doi.org/10.7554/eLife.38839.025
Author response https://doi.org/10.7554/eLife.38839.026

## Additional files

### Supplementary files

• Supplementary file 1. BioID of PCSK9-interacting proteins. Spectral counts are listed for prey proteins identified from streptavidin-purified lysates of cells expressing fusions of BirA* with GFP, PCSK9, A1AT, SAR1A, and SAR1B. Gene Ontology enrichment analysis (*Ashburner et al., 2000*; *The Gene Ontology Consortium, 2017*) is displayed for candidate PCSK9 cargo receptors.
DOI: https://doi.org/10.7554/eLife.38839.014

• Supplementary file 2. CRISPR screen sgRNA-level analysis. DESeq2 output for each sgRNA showing normalized mapped reads in control (GFP-low) populations and log2 fold-change in experiment (GFP-high) populations with significance testing for enrichment or depletion. A copy of the reference library showing the corresponding gene target and target sequence and gene for each sgRNA is included.
DOI: https://doi.org/10.7554/eLife.38839.015

• Supplementary file 3. CRISPR screen gene-level analysis. Enrichment for each sgRNA targeting the same gene was combined to generate a gene-level enrichment score using MAGeCK (*Li et al., 2014*), which is displayed in the column corresponding to the number of unique sgRNA targeting that same that demonstrated enrichment (p<0.05) by analysis with DESeq2 (*Supplementary file 2*).
DOI: https://doi.org/10.7554/eLife.38839.016

• Supplementary file 4. Oligonucleotide sequences. Primers used for CRISPR/Cas9 targeting and genotyping are displayed.
DOI: https://doi.org/10.7554/eLife.38839.017

• Source code 1. gRNA_mapping.zip
DOI: https://doi.org/10.7554/eLife.38839.018

• Transparent reporting form
DOI: https://doi.org/10.7554/eLife.38839.019

### Data availability

Proteomic data has been deposited to the MassIVE database under accession MSV000082222. Sequencing data has been uploaded to the Sequence Read Archive under accession SRP149835. Source data files are included in the supplementary information.

The following datasets were generated:

| Author(s) | Year | Dataset title | Dataset URL | Database, license, and accessibility information |
|---|---|---|---|---|
| Emmer BT, Hesketh GG, Kotnik E, Tang VT, Lascuna PJ, Xiang J, Gingras A, Chen X, Ginsburg D | 2018 | The cargo receptor SURF4 promotes the efficient cellular secretion of PCSK9 | ftp://massive.ucsd.edu/ MSV000082222 | Publicly available at MassIVE (https://massive.ucsd.edu). |
| Emmer BT, Hesketh GG, Kotnik E, Tang VT, Lascuna PJ, Xiang J, Gingras A, | 2018 | The cargo receptor SURF4 promotes the efficient cellular secretion of PCSK9 | https://www.ncbi.nlm. nih.gov/sra?term= SRP149835 | Publicly available at the NCBI Sequence Read Archive (accession no. |

Chen X, Ginsburg D

SRP149835)

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
