## [Decision Letter]

Thank you for submitting your article "The cargo receptor SURF4 promotes the efficient cellular secretion of PCSK9" for consideration by *eLife*. Your article has been reviewed by three peer reviewers, and the evaluation has been overseen by Randy Schekman as the Reviewing/Senior Editor and Reviewer #1. The following individual involved in review of your submission has agreed to reveal their identity: Charles Barlowe (Reviewer #2).

The reviewers have discussed the reviews with one another and the Reviewing Editor has drafted this decision to help you prepare a revised submission.

Summary:

Emmer et al. have used proximity labeling to identify protein candidates for the putative ER traffic receptor responsible for sorting of PCSK9 that previous work from the Ginsburg lab suggested would respond to SEC24A for sorting into COPII vesicles. Although a number of candidate ER membrane proteins were identified, proximity labeling alone appeared not to offer the spatial or temporal resolution needed to narrow the search to a small number of candidates. They then turned to a CRISPR screen with sgRNAs covering most of the human genome and searched by FACS for cells that accumulated a PCSK9-eGFP reporter in preference to an alpha 1 antitrypsin-mCherry reporter. A fairly exhaustive screen turned up the gene SURF4, a mammalian homolog of the yeast gene ERV29, previously identified as an ER traffic receptor for the precursor of the yeast alpha mating pheromone. SURF4 was also one of the ER membrane proteins identified in their biotin proximity-labeling screen. The identification of SURF4 as important in PCSK9 was confirmed in multiple independent experiments with other tagged and untagged forms of PCSK9 expressed in wt and SURF4 del mutant lines. Consistent with a role in PCSK9 traffic from the ER, SURF4 mutant cells accumulated PCSK9 as detected in lysates of whole cells at the expense of its secretion into the medium and in the ER in intact cells. SURF4 was localized to the ER by immunofluorescence and was found associated with PCSK9 by use of a cleavable crosslinking agent followed by denaturing co-immunoprecipitation.

The work is quite interesting and compelling and adds to the tiny list of such sorting receptors found in mammalian cells. It remains a mystery how the vast majority of secreted proteins are sorted efficiently in the ER, though at least some may be secreted by a receptor-independent bulk flow mechanism. However the importance of the work is diminished to some extent as the yeast equivalent protein, Erv29, was discovered many years ago and its characterization has been reported in several papers over the years. Further, SURF4 was recently reported to facilitate apoB ER traffic in HepG2 cells. Of course, there are many things the authors could have done to further characterize the role of SURF4, such as identifying the sorting signal on PCSK9 responsible for its recognition in the ER, and examining the sorting of SURF4 into COPII vesicles formed in a cell-free reaction. The yeast protein Erv29 is sorted into COPII vesicles but not all ER membrane proteins required for ER traffic accompany their cargo into COPII vesicles (e.g. Shr3). However, as this is an initial report, these more detailed characteristics could perhaps be held for a later installment. Given the limited scope of this work, it may be more appropriately considered as an Advance on the Chen et al., paper of 2013.

Essential revisions:

1) For the cross-linking immunoprecipitation experiment shown in Figure 4D, the experiment would be strengthened by immunoblotting for additional ER/Golgi membrane proteins (e.g. LMAN1, GM130) to demonstrate specificity of the co-immunoprecipitations. At a minimum, immunoblot of actin in the IPs should be shown.

2) The authors conclude that SURF4 depletion or deletion results in only a partial effect on PCSK9 secretion and thus that other, possibly redundant functions may operate. However, the sensitivity of most of their secretion experiments is limited by the long time course of accumulation measured in steady state cells or in one case, in cells induced for PCSK9 secretion over a period of hours (Figure 3F). A clearer case for a kinetic delay dependent on SURF4 would require a pulse-chase experiment conducted over the usual transit time course of secretion (10-60 min) in wt and SURF4 del mutant cells.

3) The data in Figure 5G do not show an obvious effect of SURF4 del on the conversion of endoH sens to resistant mature PCSK9. The quantified data in 5H look good but are hard to reconcile with the appearance of the bands in 5G. For example, the intensity of the endoH resistant band of mature PCSK9 in relation to the minus enzyme control looks pretty much the same for the mutant vs. the rescue sample. Perhaps the authors could show other repetitions of this gel in a figure supplement.

---

## [Author Response]

Essential revisions:1) For the cross-linking immunoprecipitation experiment shown in Figure 4D, the experiment would be strengthened by immunoblotting for additional ER/Golgi membrane proteins (e.g. LMAN1, GM130) to demonstrate specificity of the co-immunoprecipitations. At a minimum, immunoblot of actin in the IPs should be shown.

We agree and have now performed additional immunoblots of immunoprecipitated material with antibodies to LMAN1, calnexin and PDI. A high degree of specificity is seen, with no co-IP seen for either PCSK9 or SURF4 with these control ER/ERGIC markers. These data have now been added to Figure 4D, and are described in the text (subsection “SURF4 localizes to the early secretory pathway where it physically interacts with PCSK9”).

2) The authors conclude that SURF4 depletion or deletion results in only a partial effect on PCSK9 secretion and thus that other, possibly redundant functions may operate. However, the sensitivity of most of their secretion experiments is limited by the long time course of accumulation measured in steady state cells or in one case, in cells induced for PCSK9 secretion over a period of hours (Figure 3F). A clearer case for a kinetic delay dependent on SURF4 would require a pulse-chase experiment conducted over the usual transit time course of secretion (10-60 min) in wt and SURF4 del mutant cells.

To address this important point, we have now performed pulse-chase analysis, which has been added to the manuscript as requested (Figure 5—figure supplement 1 and subsection “Loss of SURF4 results in decreased PCSK9 secretion and ER accumulation of PCSK9”, first paragraph). Indeed, we found that PCSK9 secretion into the conditioned media was detectable at 30 min in SURF4 expressing cells and significantly delayed in SURF4 null cells.

3) The data in Figure 5G do not show an obvious effect of SURF4 del on the conversion of endoH sens to resistant mature PCSK9. The quantified data in 5H look good but are hard to reconcile with the appearance of the bands in 5G. For example, the intensity of the endoH resistant band of mature PCSK9 in relation to the minus enzyme control looks pretty much the same for the mutant vs. the rescue sample. Perhaps the authors could show other repetitions of this gel in a figure supplement.

We thank the reviewers for raising this point. The intensity of the endoH-resistant band is indeed unchanged in the lysate prepared from SURF4 deficient cells. It is only the endoH-sensitive band that increases, consistent with the idea that PCSK9 ER exit is impaired but that once out of the ER, further steps in trafficking proceed with normal kinetics. We have clarified our discussion of these results in the text, and have now included the primary immunoblot data for all 4 replicates in Figure 5—figure supplement 2.